# The Incidence of Adverse Events in Adults Undergoing Procedural Sedation with Propofol Administered by Non-Anesthetists: A Systematic Review and Meta-Analysis

**DOI:** 10.3390/diagnostics15101234

**Published:** 2025-05-14

**Authors:** Flavia Pigò, Matteo Gottin, Rita Conigliaro

**Affiliations:** 1Digestive Endoscopy, Azienda Ospedaliero Universitaria of Modena, 41125 Modena, Italy; 2Gastroenterology Unit, Azienda Ospedaliero Universitaria Careggi, 50134 Firenze, Italy; matteo.gottin@gmail.com

**Keywords:** procedural sedation, endoscopy, propofol

## Abstract

**Background/Objectives:** The administration of propofol without an anesthesiologist (NAAP) during endoscopic procedures is generally considered safe. However, the available data remain limited and fragmented due to legal constraints. This systematic review and meta-analysis aimed to evaluate the incidence of adverse events in adults undergoing procedural sedation with NAAP. **Methods:** A comprehensive search was conducted in three electronic databases (MEDLINE, EMBASE, and the Cochrane Library) for studies published between 2010 and 2023. Eligible studies included randomized controlled trials and observational studies that reported predefined adverse events in adult patients receiving NAAP for procedural sedation. The analysis encompassed various types of endoscopic procedures and sedation protocols, including both balanced sedation and propofol monotherapy. Clinical heterogeneity was assessed by comparing patient characteristics, sedation methods, and outcome measures across studies. A random effects model was used for the meta-analysis, with results presented as estimated incidence rates. Subgroup analyses were conducted based on the hypoxia severity, sedation approach, and procedure type. **Results:** The search yielded 2963 records, of which 73 studies met the inclusion criteria, covering a total of 967,238 procedural sedations. Hypoxia was the most frequently reported adverse event, occurring in 40‰ of cases, followed by hypotension (38‰) and bradycardia (9‰). Severe adverse events requiring emergency intervention were rare, with an incidence of 0.12‰. The subgroup analysis indicated a low occurrence (6‰) of severe desaturation (SpO_2_ < 80%) and no significant differences in adverse event rates between balanced propofol sedation and propofol-only sedation. However, advanced endoscopic procedures (EUS, ERCP, PEG, enteroscopy, EMR/ESD) were associated with a higher risk of hypoxia (10% vs. 26‰; *p* < 0.00001) and major complications (3.1‰ vs. 0.1‰; *p* = 0.015) compared to diagnostic procedures. **Conclusions:** NAAP-based procedural sedation appears to be generally safe. While the minor adverse event rates vary depending on the sedation regimen and procedure type, major complications remain exceptionally rare.

## 1. Introduction

Propofol is widely favored for both diagnostic and therapeutic endoscopic procedures due to its ease of administration, rapid onset, and short duration of effect. Although its use is typically restricted to anesthesiologists under regulations set by agencies such as AIFA and the FDA, non-anesthesiologist administration (NAAP) during endoscopic procedures has been reported to be safe. However, the data on this practice remain limited and dispersed due to legal constraints [1,2].

The growing shortage of both financial and human resources in public healthcare systems has led to the exploration of alternative sedation strategies. In this context, NAAP has emerged as a cost-effective approach [3,4]. Meta-analyses comparing propofol administration by anesthesiologists versus non-anesthesiologists suggest that both methods have similar safety profiles. Moreover, NAAP procedures often involve lower propofol doses and have been associated with higher satisfaction rates among patients and endoscopists [5,6,7].

Various retrospective and prospective studies have evaluated the safety of NAAP, consistently reporting low incidences of both minor and major adverse events [8,9,10,11]. However, a comprehensive analysis that integrates findings from both retrospective and prospective studies is still lacking. Despite methodological differences, these studies typically involve significantly larger populations than randomized controlled trials (RCTs).

Conducting a comprehensive meta-analysis aimed at summarizing the incidence of adverse events could yield critical evidence regarding the safety profile of NAAP. Such findings would provide valuable contributions to international scientific societies, supporting the integration of NAAP into clinical practice. This process would enable the development of standardized protocols to be included in clinical guidelines, thereby facilitating its adoption not only from a clinical but also a medico-legal perspective, ultimately leading to significant improvements in the cost-effectiveness, quality, and safety of endoscopic procedures.

This study aimed to perform a meta-analysis to assess the incidence of adverse events in adults undergoing procedural sedation with propofol administered by non-anesthesiologists during digestive endoscopy.

## 2. Materials and Methods

### 2.1. Study Design

This meta-analysis was conducted in accordance with the Preferred Reporting Items for Systematic Reviews and Meta-Analyses (PRISMA) guidelines and was registered in the Prospero register (ID CRD420250654453).

### 2.2. Eligibility Criteria

**Types of studies.** We aimed to include original research studies, such as randomized controlled trials (RCTs), observational studies, and retrospective analyses, that investigated the use of propofol for sedo-analgesia in adult patients undergoing endoscopic procedures, administered by non-anesthesiologist personnel. No language restrictions were applied. To minimize the variability in findings, only studies published from 2010 onward were considered for inclusion.

**Types of patients and procedures.** All pharmacological agents used for procedural sedation in adult patients were considered. The included procedures encompassed esophagogastroduodenoscopy (EGDS), colonoscopy, endoscopic retrograde cholangiopancreatography (ERCP), endoscopic ultrasound (EUS), percutaneous endoscopic gastrostomy (PEG), and enteroscopy.

**Types of interventions.** For inclusion, procedural sedation with propofol had to be administered by non-anesthesiologist personnel, including endoscopists, nurses, or nurses under the supervision of an endoscopist. Both single-agent and combination drug regimens were considered. Propofol is commonly used alone or in conjunction with other medications, such as benzodiazepines or opioids, as part of a balanced sedation approach. Patients receiving sedation were monitored in accordance with the guidelines outlined in each study’s protocol.

### 2.3. Study Protocol

**Search strategy.** A comprehensive search of three electronic databases—MEDLINE, EMBASE, and the Cochrane Library—was designed and conducted by an expert methodologist, covering the period from 2010 to 2023. The detailed search strategy can be found in Appendix A.

**Study selection.** Two investigators (FP, MG) independently reviewed all titles and abstracts to determine eligibility. Studies that were deemed potentially relevant were obtained in full text for further assessment. Any discrepancies were discussed with a third investigator (RC) and resolved through consensus.

**Data extraction.** Collected data included study design and the incidence of each reported adverse event. The information was cross-checked between the two investigators, with any discrepancies being resolved through discussion. Details on the medications that were used were documented, specifying whether a single agent or a combination was administered. For studies comparing different sedation regimens, only cases involving propofol sedation were included. If a study analyzed multiple patient cohorts (e.g., different age groups), the overall incidence of adverse events was calculated. Additionally, the total number of patients who experienced adverse events and the total number of procedures performed were recorded.

**Risk of bias assessment.** For randomized controlled trials (RCTs), we evaluated the risk of bias using the Cochrane Collaboration’s bias assessment tool [12]. For cohort studies, the risk of bias was assessed using the Newcastle–Ottawa Scale [13]. Clinical heterogeneity was examined by comparing the participant characteristics, interventions, and outcome measures to determine their similarity across the studies.

**Missing data.** Outcomes were collected as reported in the published studies, and authors were contacted via email if any data were missing or unclear. If the data remained unavailable after contacting the authors, the study was categorized as unclear. Any outcomes not reported in the study were also noted in the data extraction form.

**Variable criteria of outcomes**. The studies included in the analysis defined outcome events like hypoxia, hypotension, and bradycardia using varying criteria. We analyzed these outcomes according to the specific definitions provided in each study. For hypoxia, the following saturation categories were considered: SpO_2_ 95–91%, SpO_2_ 90–86%, SpO_2_ 85–81%, and SpO_2_ < 80%.

**Clinical outcomes.** To select the adverse events for extraction and reporting, we referred to commonly reported outcomes in the existing literature. After discussion, we reached an agreement on the following outcome measures: minor events (hypoxia, hypotension, bradycardia) and major events (death, sustained reduced awareness, permanent injury, the requirement for emergency anesthesiologist evaluation, or the need for hospitalization or endotracheal intubation).

### 2.4. Data Analysis

We used Stata 13^®^ software for the meta-analysis (metaprop function) following a random effects model as described by DerSimonian-Laird [14]. I^2^ was used to quantify the degree of statistical heterogeneity between studies [15]. We estimated the incidence per 1000 patients with a 95% confidence interval (CI). Only studies that reported the adverse events rate were used to calculate the incidence. When the number was zero, we calculated the CI using the modified Wald method [16]. To compare the incidence rate differences between balanced propofol sedation and sedation regimen with propofol only, the test of heterogeneity between subgroups was used [17]. Incidence rate differences were also calculated between diagnostic and operative procedures and between ASA I/II and ASA III/IV patients [18].

**Subgroup analysis.** Because of variation in the cutoff and definition of hypoxia in different studies, we performed a subgroup analysis for the incidence of hypoxia by oxygen saturation (SO_2_). We also performed a subgroup analysis according to the medication used, procedure type, and ASA classes.

**Sensitivity analysis.** We performed an a priori selected sensitivity analysis with inclusion of only RCTs. These results were analyzed as a separate group.

## 3. Results

### 3.1. Characteristics of Included Studies

Figure 1 illustrates the process that we used for study selection. The search strategy yielded 2981 records for review. After eliminating duplicates, 1408 potentially relevant studies remained. Following the abstract review, 136 studies were assessed for eligibility. Ultimately, 73 articles met the inclusion criteria (see Appendix A).

### 3.2. Study Characteristics

Table 1 and Appendix A provide a summary of the included studies. Among the 73 studies, 22 were randomized controlled trials (RCTs), and 51 were observational studies (33 prospective and 18 retrospective cohorts). The studies collectively involved 943,249 patients. A total of 37 studies were conducted in Europe, 9 in America, 24 in Asia, and 3 in Oceania, with none from Africa. Fifty-one studies focused on EGDS or colonoscopy procedures, while fifteen studies included EUS procedures and nineteen studies involved ERCP procedures. Other studies covered PEG (*n* = 3), enteroscopy (*n* = 9), and ESD (*n* = 4). In 31 studies (42%), propofol was used alone, while the remaining studies employed balanced sedation. In 53 studies (73%), a nurse was responsible for administering sedo-analgesia, and propofol was delivered via bolus in 54 studies (74%), with the rest using a Target-Controlled Infusion (TCI) pump. Appendix A details the definitions of adverse events used in each study.

### 3.3. Quality and Risk of Bias Assessment

The quality assessment of the randomized controlled trials is provided in Appendix A, and the quality of the cohort studies is detailed in Appendix A. All included studies involved moderate and deep sedation. The timing of outcome measurements was consistent across studies, as sedation is closely monitored during procedures. Some clinical heterogeneity was observed due to variations in sample sizes and differences in the types of endoscopic procedures performed. The majority of RCTs (14 out of 22) had a moderate risk of bias, primarily due to the absence of blinding. Among the 51 observational studies, 46 reported results by subgroups of patients or sedation regimens, and pooling the incidence of adverse events could introduce clinical heterogeneity due to differences in study design and research questions. We did not exclude studies based on quality assessments, as this could have led to the exclusion of studies reporting rare events, particularly major adverse events.

### 3.4. Outcomes

A total of 967,238 procedural sedations involving 943,249 patients were included. Table 2 presents the incidence of adverse events per 1000 sedations. Hypoxia was the most common event, occurring in 40‰ (95% CI 34–47; I^2^ = 99%), followed by hypotension, which had an incidence of 38‰ (95% CI 28–50; I^2^ = 99%), and bradycardia at 9‰ (95% CI 6–13; I^2^ = 98%). Severe adverse events that required emergency medical intervention were rare, with an incidence of 0.12‰ (95% CI 0.04–0.16; I^2^ = 27%).

### 3.5. Subgroup Analysis

#### 3.5.1. Hypoxia Classes

A total of 71 studies involving 939,830 patients reported on the outcome of hypoxia. A subgroup analysis was performed based on each study’s definition of hypoxia (see Appendix A. Some studies reported the incidence of hypoxia at different oxygen saturation cutoffs. Seven studies with a hypoxia cutoff of 91% < SpO_2_ < 95% reported an incidence of 37‰, which was similar to the cutoff of 86% < SpO_2_ < 90% (58 studies), with an incidence of 37‰, and 81% < SpO_2_ < 85% (15 studies), with an incidence of 48‰. The incidence of hypoxia with SpO_2_ ≤ 80% (7 studies) was lower, at 6‰. Appendix A shows the forest plot estimating the incidence of hypoxia by oxygen saturation categories, along with the weight of each individual study.

#### 3.5.2. Sedation Regimen

The incidence of adverse events per medication used is displayed in Appendix A. Figure 2 shows a forest plot of the incidence of hypoxia, hypotension, bradycardia, and major events with respect to the type of sedation and representation of weight of single study.

Midazolam/Propofol. The hypoxia, hypotension, and bradycardia rates were 29‰, 35‰, and 4‰. The incidence of major events was notably low (0.1‰; 95%CI 0.05–0.2) with heterogeneity I^2^ = 0%.

Midazolam/Opioid/Propofol. The hypoxia, hypotension, and bradycardia rates were 26‰, 12‰, and 0.1‰. The incidence of major events was low (0.1‰; 95%CI 0.001–0.12; I^2^ = 10%).

Opioid/Propofol. Opioid/propofol studies had the highest rate of hypoxia (87‰) but with a high degree of variability because different opioids were used (alfentanil, fentanyl, pentazocine). The incidence of hypotension was highest with opioid/propofol (96‰) for the same reason as above. In the study of Poincloux, the definition of a hypotensive episode was based on the mean arterial pressure rather than systolic pressure. The incidence of bradycardia was highest with the use of opioid/propofol (22‰). In the study of Poincloux, bradycardia events were reported but not the definition of the event. Opioid/propofol sedation regimen showed a low rate of major adverse events (1‰; 95%CI 0.05–5.0; I^2^ = 79%).

Propofol only. Procedural sedation with propofol only showed a significantly higher incidence rate of hypoxia (42) and hypotension (33‰). The bradycardia rate was 13‰. Major events were, however, low (0.1‰).

Propofol vs. balanced propofol sedation. A total of 31 studies reported sedation with only propofol on 282,713 patients, and a total of 42 studies on 660,536 patients reported adverse events for procedures with balanced propofol sedation (midazolam/propofol or midazolam/opioid/propofol or opioid/propofol). Balanced propofol sedation showed less hypoxia events (37‰) compared with a propofol-only sedation regimen but without significance (*p*-value 0.45). Procedural sedation with balanced propofol sedation showed, also if not significant, a higher incidence rate of hypotension (42‰ vs. 33‰) compared with propofol only (*p*-value 0.48). Bradycardia was reported for balanced propofol sedation with an incidence of 13‰. No differences exist in major events between balanced propofol sedation (0.1‰) and propofol (0.1‰) only (*p*-value 0.2).

#### 3.5.3. Procedure Type

A total of 19/73 studies reported adverse events for different types of procedure in the same article, so they were not considered for this subgroup analysis. Appendix A shows adverse events divided according to the type of procedure. Figure 3 shows a forest plot of the incidence of hypoxia, hypotension, bradycardia, and major events with respect to the type of procedure and representation of weight of a single study. Forest plots for PEG and enteroscopy are not displayed due to a paucity of studies.

EGDS/Colonoscopy. In total, 38 studies examined adverse events on EGDS, colonoscopies, or both in 483,681 patients. The incidence of hypoxia was 30‰, that of hypotension was 40‰, and for bradycardia, it was 10‰. Major events during EGDS and colonoscopies were 0.03‰.

Echoendoscopy. Four studies reported adverse events based on 927 patients. Hypoxia events were reported in 98 cases per 1000 exams, and the hypotension rate was 10‰. Bradycardia events were reported in 28 cases per 1000 exams. Major events occurred in 0.001‰ of cases.

ERCP. Eight studies reported adverse events based on 1937 patients. Hypoxia occurred with a rate of 86 cases per 1000 procedures. Hypotensive cases and bradycardia cases were reported with an incidence of 35‰ and 34‰, respectively. Major events were reported in 0.01‰ of cases.

PEG. Two studies based on 194 patients reported adverse events. The hypoxia rate was notably high (42.7%), because Michael et al. considered mild desaturation (SO_2_ < 90% > 15 s) and severe oxygen desaturation (SO_2_ < 85%) to be adverse events. The rates of hypotension and bradycardia cases were 42‰ and 0, but these data only refer to one study. In addition, the occurrence of major events was zero, but they were only reported only in one study.

Enteroscopy. Two studies (one on single-balloon procedures and one with double-balloon procedures) on 73 patients reported adverse events. Hypoxia occurred at a rate of 40 cases per 1000 procedures. Hypotensive cases and bradycardia cases were reported with an incidence of 68‰ and 18‰, respectively. No major events were reported.

#### 3.5.4. First-Level vs. Second-Level Procedures

A total of 38 studies only reported adverse events for first-level procedures (diagnostic EGDS/colonoscopy). In addition, 42 studies only reported adverse events for second-level procedures (EGDS/colonoscopy with EMR/ESD, EUS, ERCP, PEG, enteroscopy). Significant differences were clear in terms of hypoxia events (26‰ vs. 10%; *p*-value < 0.00001) and major events (0.1‰ vs. 3.1‰; *p*-value 0.015) for first-level and second-level procedures, respectively.

#### 3.5.5. ASA Classes

Sixty studies reported the ASA classification of patients. Overall, in the studies that classified patients according to the ASA score, adverse events were reported for 520,565 ASA I/II patients and for 83,051 ASA III/IV patients. With the exception of one study [88], conducted only on ASA III/IV patients, the rest of the studies involving ASA III/IV patients reported adverse events without distinguishing which ASA score the patients belonged to. Therefore, an incidence analysis for ASA subgroups was not performed.

#### 3.5.6. Sensitivity Analysis

A sensitivity analysis of 22 RCT studies was performed. The incidence of hypoxia (84‰) was significantly higher compared to prospective/retrospective studies (30‰; *p*-value = 0.003). Bradycardia was also higher (22‰) for RCT studies compared with prospective/retrospective studies (7‰; *p*-value = 0.039). Hypotension (36‰) and major adverse events (0.03‰) were similar to non-randomized studies (40‰ and 0.06‰, respectively). Appendix A presents an overview of these findings.

## 4. Discussion

We documented the occurrence of adverse events during procedural sedation with propofol administered by non-anesthesiologist personnel. Our analysis included 73 studies, encompassing a total of 943,249 patients. The frequency of severe adverse events requiring emergency intervention, causing permanent harm, or resulting in death was notably low (fewer than 1 event per 1000 sedations).

Endoscopic procedural sedation is beneficial in improving patient compliance, enhancing satisfaction for both patients and endoscopists, and ensuring procedural safety and efficiency.

However, the use of propofol for sedation during GI endoscopy still presents challenges, particularly in relation to legal concerns regarding who is authorized to administer sedation, as well as the increased risk of cardiopulmonary adverse events. These issues contribute to growing uncertainty regarding the feasibility of implementing NAAP in endoscopic procedures and have led to an increasing demand for the standardization of NAAP protocols. Furthermore, there is substantial variability in adherence to sedation guidelines, pre-procedural risk stratification, and the management of sedation-related complications, as recently highlighted in a European survey on this topic [89].

This meta-analysis incorporates a substantial number of patients and procedures; as expected, both the clinical and statistical heterogeneity were high. This variability is a strength and a limitation of the study. Clinical heterogeneity is represented by the differences in sedation types (balanced sedation, propofol-only boluses, propofol in TCI), procedure types (diagnostic vs. operative), patient populations (ASA I/II vs. ASA III/IV), and the classification of minor adverse events. For example, hypoxia was categorized in several ways across studies, with variations in oxygen saturation cutoffs ranging from 95% to 80%, and some studies considered transient drops in saturation for brief periods as an adverse event. While such fluctuations are common and can be managed with non-invasive interventions (e.g., chin extension, jaw thrust, or increasing oxygen flow), persistent desaturation rarely necessitates invasive interventions like mechanical ventilation. The incidence of major adverse events remained low, reinforcing the reliability of the classification of severe events across studies (I^2^ = 27%).

The effects of sedative drugs on blood pressure and heart rate were similarly influenced by variations in how adverse events were categorized (see Appendix A). Hypotension during sedation is generally transient and typically managed with fluid infusion, requiring minimal intervention. The effect on the heart rate can result from both drug effects and the physiological responses to painful stimuli during the procedure (e.g., vagal response). The clinical relevance of mild hypoxemia or hypotension is debated, with some suggesting that severe hypoxia or hypotension may contribute to ischemic changes. However, patients with pre-existing conditions are at higher risk of cardiac or cerebral damage. Furthermore, a study by Aguirre et al. highlighted a greater adverse impact on neurobehavioral function 24 h post-surgery in patients experiencing cerebral desaturation [88,90]. Therefore, thorough pre-procedural assessments of the cardiovascular and respiratory risks are essential, and most guidelines do not recommend NAAP for ASA III/IV patients [91].

We also assessed different sedation regimens and found no significant differences in the rate of serious adverse events between them. We compared 31 studies using propofol-only sedation and 42 studies using balanced propofol sedation, finding no notable differences in hypoxia, hypotension, or bradycardia events. Major adverse events were also similar between the two regimens (0.1‰ vs. 0.1‰; *p*-value = 0.2). Both NAAP regimens appear to be safe, with the choice of sedation method depending on the patient’s characteristics, the type of procedure, and the setting of the endoscopic unit. Propofol-only sedation is recommended for quick procedures, although it should be avoided in elderly patients or those with impaired left ventricular function. The main benefit of balanced sedation is the reduced overall propofol dosage, although this can result in a slower recovery post-procedure [92,93,94].

There is still uncertainty regarding NAAP for advanced endoscopic procedures, as these procedures typically require more sedation and can be complicated by factors such as abdominal distension from air insufflation, which increases the risk of cardiopulmonary events. Three meta-analyses have addressed this issue. A meta-analysis of 26 prospective observational studies compared propofol sedation administered by non-anesthesia providers with that provided by anesthesia providers in over 5000 advanced procedures (EUS, ERCP, and small-intestinal enteroscopy). NAAP was associated with similar safety outcomes, although it led to lower satisfaction levels among both patients and endoscopists compared to sedation provided by anesthesia providers. Two other meta-analyses compared propofol sedation to traditional sedation methods for advanced procedures in general (nine RCTs, 969 patients) and specifically for ERCP (six RCTs, 663 patients). Both studies found that propofol sedation was associated with shorter recovery times and comparable cardiopulmonary adverse events; additionally, the largest study indicated better sedation and amnesia outcomes with propofol [5,6,7].

In our analysis, 38 studies only reported adverse events for first-level procedures (diagnostic EGDS/colonoscopy), while 42 studies focused on second-level procedures (EGDS/colonoscopy with EMR/ESD, EUS, ERCP, PEG, enteroscopy). Significant differences were found in hypoxia events (26‰ vs. 10‰; *p*-value < 0.00001) and major events (0.1‰ vs. 3.1‰; *p*-value = 0.015) for first- and second-level procedures, respectively. These differences, compared to other meta-analyses, may stem from a larger number of studies, including more recent data [95,96,97]. Although NAAP appears to be relatively safe even in advanced procedures, further studies are needed to thoroughly assess its safety profile in this setting. Its use may be more easily standardized in complex procedures performed by a more selected and experienced group of operators, typically within a limited number of specialized centers. However, the greater procedural complexity, longer operative times, and need for increased intra-procedural assistance must be considered, especially in situations where dedicated staff for the exclusive administration and monitoring of sedation may not always be available.

Our sensitivity analysis showed a higher incidence of hypoxia and bradycardia in RCTs compared to non-randomized studies, suggesting that minor adverse events might be overestimated in randomized trials. However, the incidence of major adverse events remained consistently low, indicating that severe complications are less influenced by the variability in definitions and potential recall bias.

To further reduce adverse events, standardization of procedures is essential across all endoscopic units. Several clinical recommendations advocate for education and training, such as the European Curriculum for Sedation, which sets guidelines for training non-anesthesiologists (physicians and nurses) who will administer sedation during gastrointestinal endoscopies. This course integrates theoretical and practical knowledge and covers various topics, including anatomy, pharmacology, patient risk assessment, sedation management, and guidelines for sedation complications [98,99,100,101].

## 5. Conclusions

In conclusion, our meta-analysis strengthens the existing evidence that NAAP is a safe sedation strategy for first-level diagnostic procedures when administered as balanced or propofol-only sedation. The implementation of standardized protocols in endoscopic units, accounting for patient- and procedure-related risk factors, as well as proper training of personnel involved in NAAP administration, is essential to ensure effective and safe sedation. Moreover, such an approach could facilitate the integration of NAAP into routine clinical practice, yielding significant benefits in terms of procedural quality, patient satisfaction, and healthcare cost-effectiveness.

## Figures and Tables

**Figure 1 diagnostics-15-01234-f001:**
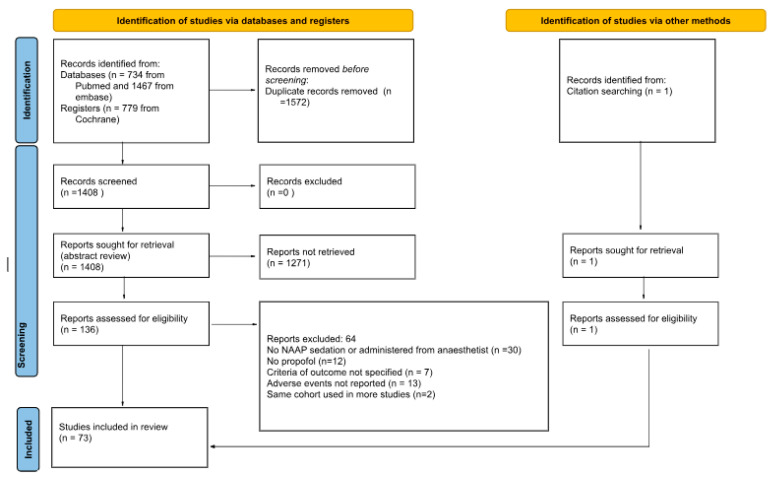
Study selection process.

**Figure 2 diagnostics-15-01234-f002:**
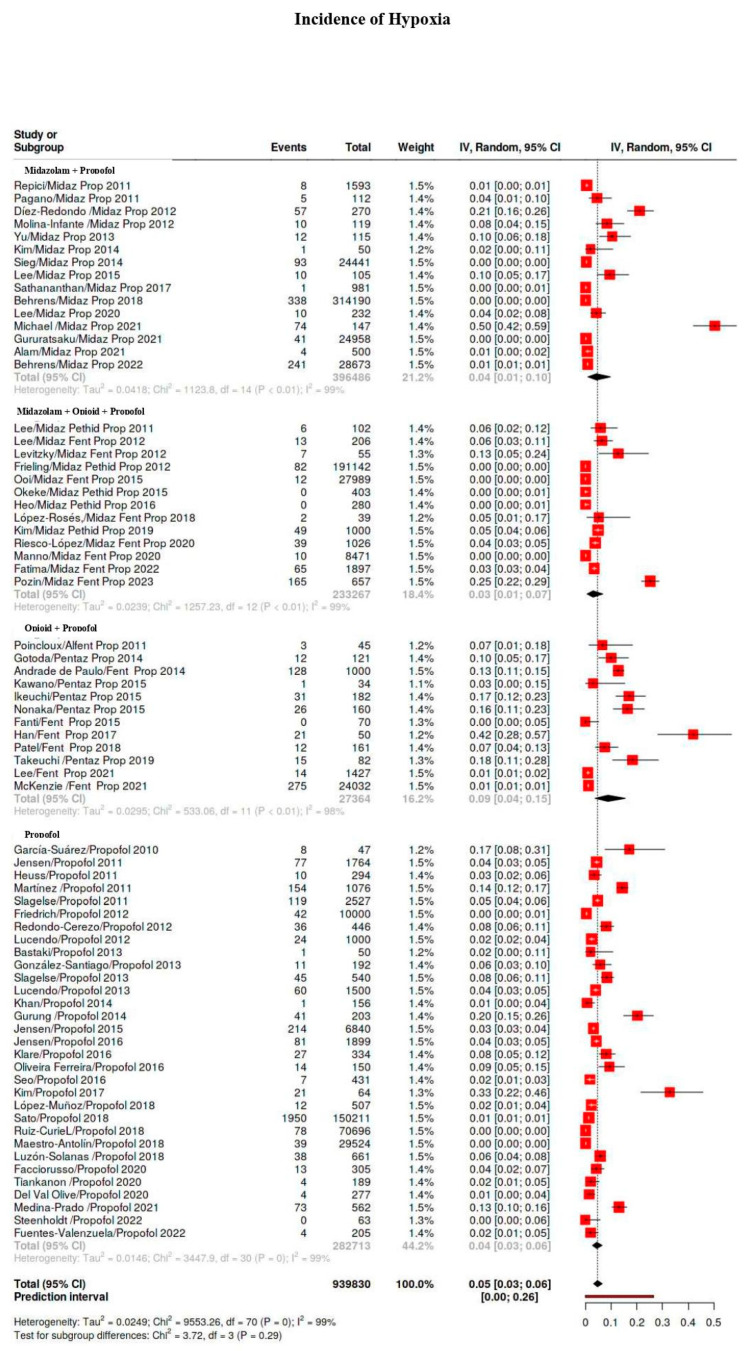
Forest plot of incidence of hypoxia, hypotension, bradycardia, and major adverse events with respect to type of sedation and representation of weight of single study [8,9,10,11,19,20,21,22,23,24,25,26,27,28,29,30,31,32,33,34,35,36,37,38,39,40,41,42,43,44,45,46,47,48,49,50,51,52,53,54,55,56,57,58,59,60,61,62,63,64,65,66,67,68,69,70,71,72,73,74,75,76,77,78,79,80,81,82,83,84,85,86,87].

**Figure 3 diagnostics-15-01234-f003:**
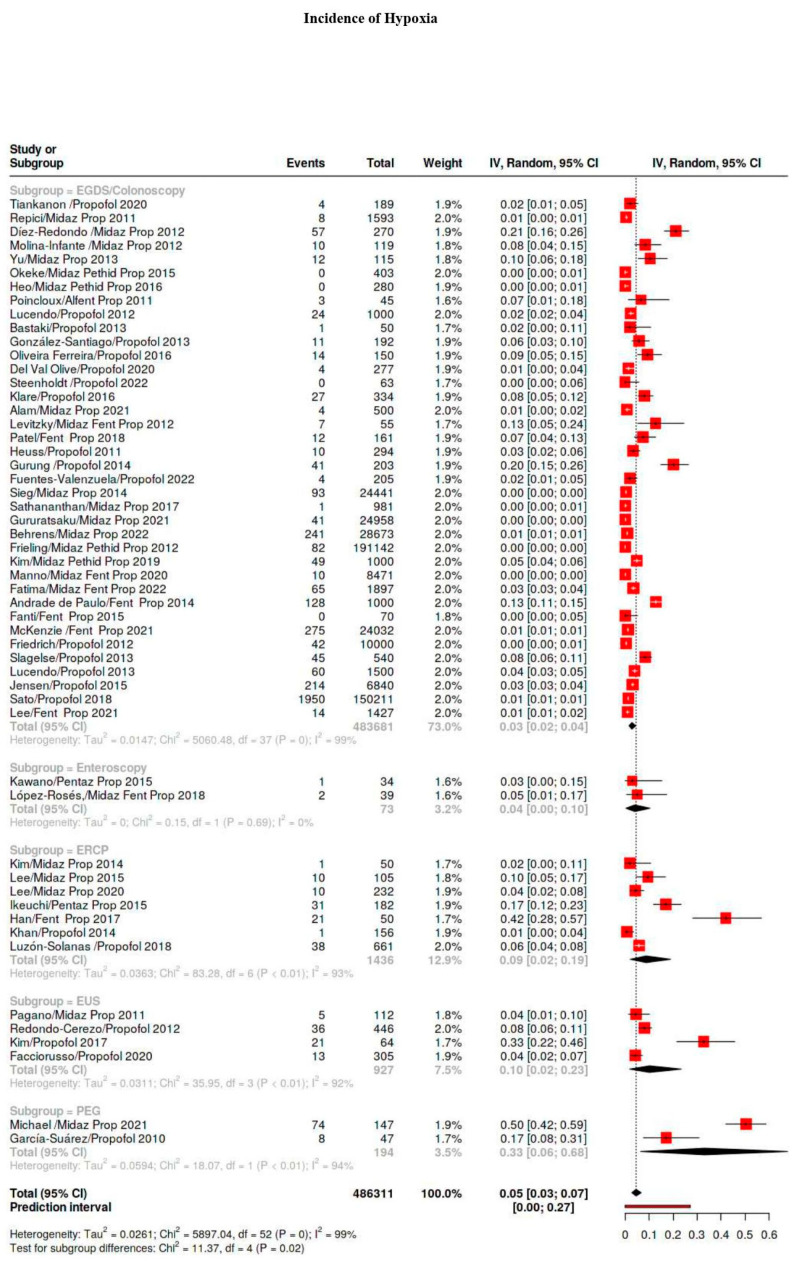
Forest plot of incidence of hypoxia, hypotension, bradycardia, and major adverse events with respect to type of procedure and representation of weight of single study [8,9,10,11,19,20,21,22,23,24,25,26,27,28,29,30,31,32,33,34,35,36,37,38,39,40,41,42,43,44,45,46,47,48,49,50,51,52,53,54,55,56,57,58,59,60,61,62,63,64,65,66,67,68,69,70,71,72,73,74,75,76,77,78,79,80,81,82,83,84,85,86,87].

**Table 1 diagnostics-15-01234-t001:** Characteristics of the included studies.

Study	Year	Design	No. Patients	Procedures	Medication Used	Administrator	Way of Administration	ASA
Akyuz [19]	2010	Retro	2918	EGDS Colon ERCP DBE	Mida Propofol	NuES	Bolus	I–IV
García-Suárez [20]	2010	Prosp	47	PEG	Propofol	NuES	Bolus	III–IV
Poincloux [21]	2011	RCT	45	Colon	Alfent Propofol	Endoscopist	Bolus	I/II
Repici [22]	2011	Prosp	1593	Colon	Mida Propofol	Endoscopist	Bolus	I/II
Lee [23]	2011	RCT	102	EGDS ERCP	Mida Pethid Propofol	Nurse	Bolus	I–IV
Pagano [24]	2011	Prosp	112	EUS	Mida Propofol	Endoscopist	Bolus	I/II
Jensen [25]	2011	Prosp	1764	EGDS Colon EUS ERCP DBE	Propofol	NuES	Bolus	I–IV
Heuss [26]	2011	RCT	294	EGDS	Propofol	Nurse	Bolus	I–IV
Martínez [27]	2011	Prosp	1076	EGDS Colon EUS	Propofol	Endoscopist	TCI	I–IV
Slagelse [28]	2011	Prosp	2527	EGDS Colon EUS ERCP DBE	Propofol	Nurse	Bolus	I–IV
Lee [29]	2012	RCT	206	EUS ERCP	Mida Fent Propofol	NuES	Bolus	I–IV
Díez-Redondo [30]	2012	RCT	270	Colon	Mida Propofol	Endoscopist	Bolus	NA
Friedrich [31]	2012	Prosp	10,000	EGDS Colon	Propofol	Nurse or Endoscopist	Bolus	NA
Redondo-Cerezo [32]	2012	Prosp	446	EUS	Propofol	NuES	Bolus	I–IV
Levitzky [33]	2012	RCT	55	EGDS	Mida Fent Propofol	Endoscopist	Bolus	I–IV
Lucendo [34]	2012	Prosp	1000	Colon	Propofol	NuES	Bolus	I/II
Molina-Infante [35]	2012	RCT	119	Colon	Mida Propofol	NuES	Bolus	I–IV
Frieling [8]	2012	Prosp	191,142	EGDS Colon	Mida Pethid Propofol	Nurse or Endoscopist	Bolus	NA
Bastaki [36]	2013	RCT	50	Colon	Propofol	Nurse	Bolus	I–II
González-Santiago [37]	2013	RCT	192	Colon	Propofol	Nurse	Bolus/TCI	I–IV
Slagelse [38]	2013	RCT	540	EGDS Colon	Propofol	Nurse	Bolus	I–IV
Lucendo [39]	2013	Prosp	1500	EGDS Colon	Propofol	NuES	Bolus	I–IV
Yu [40]	2013	RCT	115	Colon	Mida Propofol	NuES	Bolus	I–IV
Kim [41]	2014	Retro	50	ERCP	Mida Propofol	NuES	Bolus	I/II
Gotoda [42]	2014	Retro	121	EGDS (ESD)	Pentaz Propofol	Endoscopist	TCI	I–IV
Sieg [43]	2014	Prosp	24,441	EGDS Colon	Mida Propofol	NuES	Bolus	NA
Khan [44]	2014	Prosp	156	ERCP	Propofol	Endoscopist	Bolus	I–IV
Gurung [45]	2014	Prosp	203	EGDS	Propofol	NuES	Bolus	NA
Andrade de Paulo [46]	2014	Prosp	1000	EGDS Colon	Fent Propofol	Nurse	Bolus	I–II
Kawano [47]	2015	Prosp	34	Entero	Penthaz Propofol	Endoscopist	TCI	NA
Lee [48]	2015	RCT	105	ERCP	Mida Propofol	NuES	Bolus	I–IV
Ikeuchi [49]	2015	Prosp	182	ERCP	Penthaz Propofol	Endoscopist	Bolus/TCI	I–IV
Jensen [50]	2015	Retro	6840	EGDS Colon	Propofol	NuEs	Bolus	I–IV
Ooi [11]	2015	Prosp	27,989	EGDS Colon PEG	Mida Fent Propofol	NuES	Bolus	NA
Nonaka [51]	2015	Prosp	160	ERCP EGDS (ESD)	Pentaz Propofol	Endoscopist	Bolus	I–IV
Fanti [52]	2015	RCT	70	EGDS Colon	Fent Propofol	Endoscopist	TCI	I/II
Okeke [53]	2015	Retro	403	Colon	Mida Pethid Propofol	Endoscopist	Bolus	NA
Heo [54]	2016	RCT	280	Colon	Mida Pethid Propofol	Nurse	Bolus	I–IV
Jensen [55]	2016	Prosp	1899	ERCP EUS DBE	Propofol	Nurse	Bolus	I–IV
Klare [56]	2016	RCT	334	Colon	Propofol	Endoscopist	Bolus	I–IV
Oliveira Ferreira [57]	2016	RCT	150	Colon	Propofol	NuES	Bolus	I/II
Seo [58]	2016	Retro	431	EGDS Colon (EMR/ESD)	Propofol	NuES	Bolus/TCI	I–IV
Sathananthan [59]	2017	Prosp	981	EGDS Colon	Mida Propofol	NuES	Bolus	I–IV
Han [60]	2017	RCT	50	ERCP	Fent Propofol	Nurse	Bolus	I–IV
Kim [61]	2017	RCT	64	EUS	Propofol	NuES	Bolus	I/II
Behrens [62]	2018	Prosp	314,190	EGDS Colon EUS ERCP Entero	Mida Propofol	Nurse or Endoscopist	NA	I–IV
López-Muñoz [63]	2018	Prosp	507	EGDS Colon ERCP EUSDBE	Propofol	NuES	NA	NA
Sato [10]	2018	Prosp	150,211	EGDS Colon	Propofol	Nurse	Bolus	I/II
Patel [64]	2018	Retro	161	EGDS	Fent Propofol	NuES	Bolus	I–IV
Ruiz-Curiel [9]	2018	Retro	70,696	EGDS Colon ERCP EUS	Propofol	NuES	Bolus	NA
Maestro-Antolín [65]	2018	Retro	29,524	EGDS Colon ERCP EUS	Propofol	Endoscopist	Bolus/TCI	I–IV
Luzón-Solanas [66]	2018	Prosp	661	ERCP	Propofol	Endoscopist	TCI	I–IV
López-Rosés [67]	2018	Prosp	39	Entero	Mida Fent Propofol	NuES	TCI	I–IV
Kim [68]	2019	Retro	1000	EGDS Colon	Mida Pethid Propofol	Nurse or Endoscopist	Bolus	I/II
Takeuchi [69]	2019	Retro	82	EGDS (ESD)	Pentaz Propofol	Endoscopist	TCI	I–IV
Lapidus [70]	2019	Retro	501	ERCP	Mida Fent Propofol	Endoscopist	Bolus	I–IV
Lee [71]	2020	RCT	232	ERCP	Mida Propofol	NuES	Bolus/TCI	NA
Facciorusso [72]	2020	Prosp	305	EUS	Propofol	Nurse or Endoscopist	Bolus	I–IV
Riesco-López [73]	2020	Prosp	1026	EGDS Colon EUS	Mida Fent Propofol	NuES	Bolus	I–IV
Tiankanon [74]	2020	Retro	189	Colon	Propofol	Nurse	TCI	I/II
Del Val Oliver [75]	2020	Retro	277	Colon	Propofol	NuES	NA	I/II
Manno [76]	2020	Prosp	8471	EGDS Colon	Mida Fent Propofol	NuES	Bolus	I–IV
Michael [77]	2021	RCT	147	PEG	Mida Propofol	NuES	Bolus	NA
Lee [78]	2021	Retro	1427	EGDS Colon	Fent Propofol	NuES	TCI	I–IV
Gururatsaku [79]	2021	Prosp	24,958	EGDS Colon	Mida Propofol	Nurse	Bolus	I–IV
Alam [80]	2021	Prosp	500	EGDS	Mida Propofol	Nurse or Endoscopist	Bolus	I–IV
Medina-Prado [81]	2021	Prosp	562	EGDS Colon EUS	Propofol	Endoscopist	TCI	I–IV
McKenzie [82]	2021	Retro	24,032	EGDS Colon	Fent Propofol	NuES	Bolus	I–IV
Steenholdt [83]	2022	RCT	63	Colon	Propofol	NuES	Bolus	I–II
Fuentes-Valenzuela [84]	2022	Prosp	205	EGDS	Propofol	Endoscopist	NA	NA
Behrens [85]	2022	RCT	28,673	EGDS Colon	Mida Propofol	Nurse or Endoscopist	Bolus	I/II
Fatima [86]	2022	Retro	1897	EGDS Colon	Mida Fent Propofol	NuES	Bolus	I–IV
Pozin [87]	2023	Retro	657	EGDS Colon EUS ERCP DBE	Mida Fent Propofol	NuES	Bolus	

Prosp (prospective observational study); Retro (retrospective); RCT (randomized controlled trial); EGDS (esophagogastroduodenoscopy); Colon (colonoscopy); EUS (endoscopic ultrasound); ERCP (endoscopic retrograde colangiopancreatography); PEG (percutaneous endoscopic gastrostomy); Entero (enteroscopy); DBE (double-balloon enteroscopy); ESD (endoscopic submucosal dissection); EMR (endoscopic mucosal resection); Mida (midazolam); Fent (fentanyl); Pethid (Pethidine); Penthaz (Penthazocine); NuES (nurse under endoscopist supervision); TCI (Target-Controlled Infusion); NA (not assessed).

**Table 2 diagnostics-15-01234-t002:** Incidence of adverse events per 1000 procedural sedations.

Adverse Event	Events per Patients	Estimate per 1000	95%CI	I^2^ (%)
Hypoxia	5101/939,830	40	34–47	99
Hypotension	5329/569,506	38	28–50	99
Bradicardia	1586/641,934	9	6–13	98
Major	109/941,562	0.12	0.04–0.16	27

Results are presented as the number of events over the total of patients (only studies that reported the events), estimate based on 1000 patients, 95% confidence interval, and heterogeneity index (I^2^).

## Data Availability

The data presented in this study are available on request from the corresponding author. The data are not publicly available due to privacy/ethical restrictions.

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
