# Peer review of "The Incidence of Adverse Events in Adults Undergoing Procedural Sedation with Propofol Administered by Non-Anesthetists: A Systematic Review and Meta-Analysis"

_diagnostics, 2025, doi:10.3390/diagnostics15101234_

Round 1
Reviewer 1 Report
Comments and Suggestions for Authors
Thank you for the opportunity to review this manuscript. The use of propofol sedation in endoscopic procedures is highly beneficial, and the meta-analysis presented in this paper is well-organized and clearly written, which I believe will engage a high level of interest among readers. To enhance reader comprehension and to maximize the impact of the research findings, I would appreciate if the following points could be addressed:
- Given the large number of studies analyzed, including a table within the main text detailing the characteristics of these studies would likely be of great interest to readers.
- The conclusion notes that while NAAP is preferable for diagnostic endoscopic procedures due to fewer complications, the practical challenges of deploying NAAP, such as the high volume of procedures and limited trained personnel, make it difficult for trained individuals to perform these roles alone. It may be more relevant to highlight the possibility of safely using NAAP without an anesthesiologist for more complex endoscopic procedures, considering the very low rate of major events. Discussing this further in the conclusion or discussion sections could be valuable.
- Figures 2 and 3, which span across large pages, would be easier to interpret if each figure included a legend. For example, in Figure 2, rather than labeling part (a) simply as "(a)", it could be labeled as "(a) Hypoxia" and instead of "Subgroup1", it could be "Subgroup1 (midazolam + propofol)" for clarity.
On page 5, line 195, please make the following corrections:
- Change "95%<SpO2<91%" to "91%<SpO2<95%".
- Change "90%<SpO2<86%" to "86%<SpO2<90%".
Reviewer 2 Report
Comments and Suggestions for Authors
The study is well-conducted and adds value to the literature, but the introduction and conclusions need slight refinement to better contextualize findings and implications. The English language is acceptable. Suggest that the authors submit a PRISMA flowchart as supplementary material. Suggest a brief comparison with recent guidelines (e.g., ESGE 2023) in the discussion.
Round 2
Reviewer 1 Report
Comments and Suggestions for Authors
Thank you very much for the opportunity to review this manuscript.
I have read the revised version and confirmed that the points I previously raised have been appropriately addressed. I have no further comments, and I believe the manuscript is suitable for acceptance.